# Preparation and Composition Optimization of PEO:MC Polymer Blend Films to Enhance Electrical Conductivity

**DOI:** 10.3390/polym11050853

**Published:** 2019-05-10

**Authors:** Hawzhin T. Ahmed, Omed Gh. Abdullah

**Affiliations:** 1Department of General Science, College of Education and Language, University of Charmo, Chamchamal 46023, Kurdistan Region, Iraq; hawzhin.taha@charmouniversity.org; 2Advanced Materials Research Laboratory, Department of Physics, College of Science, University of Sulaimani, Sulaimani 46001, Kurdistan Region, Iraq; 3Komar Research Center, Komar University of Science and Technology, Sulaimani 46001, Kurdistan Region, Iraq

**Keywords:** polymer blend films, crystallinity, optical micrographs, impedance spectroscopy, electrical modulus

## Abstract

The polymer blend technique was used to improve amorphous phases of a semicrystalline polymer. A series of solid polymer blend films based on polyethylene oxide (PEO) and methylcellulose (MC) were prepared using the solution cast technique. X-ray diffraction (XRD), Polarized optical microscope (POM), Fourier transform infrared (FTIR) and electrical impedance spectroscopy (EIS) were used to characterize the prepared blend films. The XRD and POM studies indicated that all polymer blend films are semicrystalline in nature, and the lowest degree of crystallinity was obtained for PEO:MC polymer blend film with a weight ratio of 60:40. The FTIR spectroscopy was used to identify the chemical structure of samples and examine the interactions between chains of the two polymers. The interaction between PEO and MC is evidenced from the shift of infrared absorption bands. The DC conductivity of the films at different temperatures revealed that the highest conductivity 6.55 × 10^−9^ S/cm at ambient temperature was achieved for the blend sample with the lowest degree of crystallinity and reach to 26.67 × 10^−6^ S/cm at 373 K. The conductivity relaxation process and the charge transport through the hopping mechanism have been explained by electric modulus analysis. The imaginary part of electrical modulus *M*″ shows an asymmetrical peak, suggesting a temperature-dependent non-Debye relaxation for the PEO:MC polymer blend system.

## 1. Introduction

Polymeric materials have been receiving great research attention due to their advantages, such as light-weight, low-cost, high flexibility, good mechanical properties, and ease of fabrication in thin film form [1]. These properties make them ideal materials for a broad range of applications in optical, biomedical, and electronic devices [2]. It has been established that the features of the polymeric materials depend mainly on the composition and attachment of monomers; therefore, many research groups have attempted to manipulate the characteristics of polymers to create materials with specific chemical, physical, and biological properties for a particular application [3]. Currently, the research interests have been focused on the solid polymer electrode-based films, because of the possible application in organic solar cells, sensors, and electrochemical devices [4,5,6].

Investigations on solid polymer electrolytes have focused mainly on the improvement of the ionic conductivities at ambient temperature [7]. However, polymeric materials possess some restrictions such as unsatisfactory mechanical properties, inferior thermal stability, and degradation at low temperature [8]. To overcome these limitations, some strategies have been employed, such as copolymerization, chemical modifications (grafting) and physical mixture (blending), plasticization, and the addition of micro/nanofillers [9,10].

In order to obtain a variety of chemical and physical properties from the constituent polymers, the blending of the polymers has been used as an economical technique, and the gain of new properties depends mainly on the degree of compatibility or miscibility of the polymer pairs at the molecular level [11]. In miscible polymer blends, the particular interactions between functional groups in the polymer segments caused a decrease of the Gibbs energy of mixture [12]. The final properties of a polymer blend commonly depend on the properties of the constituent polymers and their composition, as well as the miscibility of the polymer blend components. The main advantages of polymer blending are simplicity in preparation, economically low cost, and easy control of physical properties by compositional change [13,14].

Among the numerous reported studies of polymer blend electrolytes, blends of poly(ethylene oxide) (PEO) with an amorphous polymer have attracted academic interest. The ion conductivity in solid polymer electrolyte-based PEO is believed to take place in the amorphous phase, while the crystalline phase is insulating [15,16]; thus, by blending PEO with an amorphous polymer, the degree of crystallinity is reduced, and the conductivity of the PEO-based polymer electrolyte can be enhanced. The degree of crystallinity of PEO upon blending is dependent on the concentration of the amorphous polymers, such as methylcellulose (MC), starch, or poly(vinyl chloride) (PVC) [12,17].

PEO is a semicrystalline synthetic polymer and has the general molecular structure (CH_2_CH_2_O)*_n_*. PEO, as a commercially noticeable thermoplastic polymer, has broad application in many fields due to its high crystallinity, water-solubility, biocompatibility, and low toxicity [18]. In addition, the flexible structure of C–O–C stretching in the backbone of the PEO chain provides the material with excellent toughness [19]. On the other hand, MC is particularly interesting due to its abundance in nature, very low cost, water-solubility, nontoxicity, biocompatibility and biodegradability [20,21]. MC is a water-soluble long-chain substituted cellulose with excellent film-forming characteristics [22]. According to its degree of substitution, MC has a wide variety of uses in industry [23].

More research efforts have been directed toward the attainment of polymeric films containing large and stable amorphous phases at ambient temperature in order to increase the flexibility of the polymer chains which are responsible for the ion conduction in solid polymer electrolyte systems [24]. Polymer blending can avoid the difficulties related to the low ionic conductivity of solid polymer electrolyte materials with properties that cannot be attained with the use of a single polymer [25]. Kadir et al. [26] reported that the degree crystallinity of PEO decreases as chitosan content increases to 60 wt.% in the PEO–Chitosan blend. Ramly et al. [27] reported that a PEO–Starch blend with a ratio of 3:7 exhibits minimal crystallinity; thus, this composition was chosen in the preparation of polymer electrolyte films. This work is devoted to enhancing the conductivity of the PEO:MC blend-based polymer, by enhancing the formation of amorphous phases. Therefore, the structural, morphological, and electrical properties of different weight ratios of PEO:MC polymer blend films have been investigated, and the composition for this system has been optimized.

## 2. Experimental Details

### 2.1. Sample Preparation

In this article, no additional purification has been done for the chemicals which have been used throughout the experiment. Poly(ethylene oxide) (PEO) with molecular weight 10^6^ g/mol was purchased from Alfa Aesar (Shanghai, China), and methylcellulose (MC) of molecular weight 14,000 g/mol was provided by Merck KGaA (Darmstadt, Germany) and used as raw materials to prepare the PEO:MC blend films with different composition ratios (100:0, 80:20, 60:40, 40:60, 20:80, and 0:100) based on the solution cast technique. In the present work, 2 g of PEO was dissolved in 120 mL of distilled water, and 2 g of MC was dissolved in 240 mL of distilled water separately, at room temperature using magnetic stirrer. The two solutions are stirred continuously overnight to obtain homogeneous viscous solutions. After completely dissolving the two polymers, the obtained solutions were mixed together in the desired weight ratio under stirring for 30 min, to achieve completely homogeneous solutions. Finally, the obtained solutions were poured into different plastic Petri dishes, placed in a dust free chamber with silica gel, and allowed to dry at ambient temperature for the film to form. Prior to characterization, the films were kept in a desiccator with blue silica gel for further drying.

### 2.2. Sample Characterization

X-ray diffraction patterns of PEO:MC polymer blend films have been obtained using X’Pert Pro diffractometer (PanAnalytical, Almelo, The Netherlands) at the fixed operating voltage and current of 45 kV and 40 mA, respectively. The samples were scanned by Cu-kα1 monochromatic beam, having the wavelength 1.5406 Å, and the glancing angles were arranged in the range of 10°≤2θ≤70°. Surface morphologies of pure polymers and different compositions of the PEO:MC polymer blends were examined by digital polarized optical microscopy (AmScope 3.7 (Irvine, CA, USA)) under 10× magnification. The images were taken with software M41403 (USB 2.0). 

The Fourier transform infrared (FTIR) spectrum was recorded using a PerkinElmer Frontier spectrometer (Waltham, MA, USA). The molecular spectroscopic properties of the samples were analyzed in the wavenumber range of 400–4000 ${\mathrm{cm}}^{-1}$ with spectral resolution of 1 ${\mathrm{cm}}^{-1}$. The electrical properties of the samples were measured using a precision LCR Meter (KEYSIGHT E4980A, Agilent Inc., Santa Rosa, CA, USA) in the frequencies ranging from 100 Hz to 2 MHz, and in the temperature ranges between 303–373 K. Two aluminum electrodes with diameter of 2 cm were used as blocking electrodes for this study. The samples were inserted between them after cutting the samples to suitable shape and size. The measurements were carried out in a temperature-controlled closed chamber, and the temperature was measured using T-type thermocouple. The impedance data were presented in Argand plot, in which the sample thickness was smaller than 150 µm.

## 3. Results and Discussion

### 3.1. XRD Analysis

Figure 1 shows the X-ray diffraction (XRD) patterns of PEO, MC, and PEO:MC blend films. It can be observed that the XRD pattern of pure PEO (Figure 1a) has two sharp Bragg peaks at 19.15° and 23.35°, which indicates the semicrystalline nature of this polymer [28]. The XRD pattern of pure MC (Figure 1f) shows a broad hump around 20.15°, which, according to the literature, indicates an amorphous phase with some order in the intermolecular structure of MC [23].

From Figure 1b–e, it can be seen that with incorporation of different concentrations of MC in PEO to form blending, the intensity of the semicrystalline peaks of PEO decreased, indicating the increase in the amorphous phase due to the destruction of the ordered arrangements of the PEO chains. The amorphous enhancement in the PEO:MC polymer blends system will affect the conductivity of the samples. The amorphous phase obtained causes a reduction in the energy barrier of polymer chain segmental motion. Thus, the conductivity increases with an increase in the amorphous domain of the sample [29]. It can be predicted that the specimen with the maximum amorphous domain exhibits the highest electrical conductivity at room temperature [30,31]. To illustrate this enhancement, electrical conductivity analysis was performed on the samples.

This result demonstrates that the PEO:MC polymer blend films shows two-phase morphology, i.e., crystalline and amorphous states. To estimate the crystallinity for all samples and resolve the crystalline peaks, a combination of the scattered intensities can be used. In this method, the percentage of the degree of crystallinity $\left(X_{c}\right)$ was obtained from the ratios of the area under the crystalline peak and the corresponding amorphous halos. In order to separate the crystalline and amorphous peaks, the XRD pattern for all polymer blend compositions are deconvoluted using Fityk software (1.3.1, Warsaw, Poland, 2010) [32], and according to the following equation [33].
(1)$$X_{c}=\frac{A_{c}}{A_{c}+A_{a}}\times 100\%$$
where *A*_c_ and *A*_a_ are, respectively, the area under crystalline peaks and amorphous halos. Table 1 shows the center and the full width at half maximum (FWHM) for deconvoluted XRD patterns into Gaussian components for PEO:MC polymer blend films. Figure 2 depicts deconvoluted and fitted XRD patterns for PEO:MC polymer blend films. It can be observed that the semicrystalline peaks are deconvoluted into four or five Gaussian peaks. The degrees of crystallinity of the films are calculated from the area under the deconvoluted peaks and are tabulated in Table 1. It is observed that the $X_{c}$ value decreases gradually with increasing MC concentration until 40 wt.%, and then increases with further increasing MC concentration, achieving a minimum value of approximately 15.864%. It is clear that the PEO:MC polymer blend with a weight ratio of 60:40 exhibits a significant reduction in the crystalline phase, i.e., this composition has higher amorphicity. 

### 3.2. Surface Morphology Analysis

Polarized optical microscopy can be used to observe the spherulitic and crystalline morphology of polymers during crystallization [34]. Figure 3 shows the optical micrographs of the uncomplexed PEO, and different compositions of PEO:MC polymer blend films. The micrograph of pure PEO (Figure 3a) shows well-defined spherulitic crystalline structures separated by dark amorphous regions, which indicate the semicrystalline nature of the PEO polymer [28]. The blending of PEO with MC caused the spherulitic structures of PEO to be less prominent due to the lower radial growth rate of PEO spherulites, and finally to disappear for the pure MC sample (Figure 3f). Therefore, blending reduces the crystallinity of PEO, and thus the amorphous phase region of the polymer blend systems gradually enhanced due to a decrease in the size of spherulites, as observed from the appearance of dark regions in the blending films. Lowest crystallization rate was observed for PEO:MC with composition ratio 60:40. This result is in excellent agreement with the XRD results shown in Figure 1.

### 3.3. FTIR Spectroscopy Analysis

The chemical composition and the possible interactions between the functional groups in PEO:MC polymer blend films have been achieved by using FTIR spectroscopy. The complexation occurs between two polymers due to specific intramolecular and intermolecular interactions of polymer chains. The observed changes in the position, shape, and intensity of the IR absorption bands were used as a tool to detect all the materials interacting with each other [25]. Figure 4 shows the recorded FTIR spectra for all samples in the wavenumber range from 400 to 4000 cm^−1^.

In the case of pure PEO, for the distinctive peaks between 800–1400 cm^−1^ in the spectrum, the stretching vibration peak of C–O–C splits into three peaks at 1146, 1110, and 1062 cm^−1^, and the two bands appeared at 1344, and 1359 cm^−1^ can be ascribed to the bending vibration of CH_2_ [35]. These results reveal the existence of high crystallinity in the PEO structure [36]. However, when different amounts of MC were added to the PEO matrix, the triplet band corresponding to C–O–C stretching vibrations become broader and nearly combine into a single peak at 1067 cm^−1^, indicating a reduction of PEO crystallinity as shown in Figure 5 [35]. This observation agrees well with the results of XRD studies. From Figure 4, it can also be noted that the O–C–O bending peaks in pure PEO at 529 cm^−1^ will disappear with increasing MC; this is another indication of the existence of complexation between PEO and MC molecules [37].

Figure 6 shows that the C–H aliphatic stretching band for PEO appears at 2889 cm^−1^, and this peak was shifted to lower wavenumbers upon blending with different ratios of MC. The observed changes in the position and intensity of the C–H band reveal the change in bond length; hence, complexation takes place between the chains of the two polymers [25,26].

The hydroxyl bands (O–H) in the IR spectrum of pure PEO film, pure MC, and their blend films are shown in Figure 6. The O–H vibrational mode for pure PEO and pure MC are situated at 3520 and 3448 cm^−1^, respectively. The position of O–H band of two polymers is comparable with the previous works [20,38]. The O–H band of PEO:MC polymer blend films has shifted to 3436, 3437, 3434, 3448 cm^−1^ for weight ratios 80:20, 60:40, 40:60, and 20:80, respectively. The change in the position and intensity of both O–H and C–H band indicate the formation of hydrogen bonds between the hydrogen atoms in PEO and oxygen atom in MC, or vice versa.

### 3.4. Impedance Analysis

Impedance spectroscopy is a relatively new, nondestructive, and powerful technique for characterizing the electrical properties of electrolyte materials and their interfaces with conducting electrodes. The complex impedance planes ($Z^{\prime \prime }$ vs. $Z^{\prime }$) of the PEO:MC blend samples at different temperatures are shown in Figure 7, which consist of a single semicircle arc starting from the origin and inclined at different angles to the real axis and the frequency increases from right to left on the arcs. In general, the impedance plots exhibit two noticeable regions: the high-frequency semicircular arc and low frequency tail. For a perfect Debye-type relaxation, a complete semicircle with its center overlapping with the $Z^{\prime }$ axis should be observed. The deviation from Debye-type behaviour was expected when detecting a depressed semicircle whose center is below the $Z^{\prime }$ axis [11]. Thus, these plots separate in two regions, which correspond to the electrode polarization effect in the low frequency regions and the bulk material properties in the high frequency regions [39,40].

The nature of dielectric material in impedance spectra provides significant information about the contribution of electrode polarization phenomena (EP) (formation of electric double layer between electrodes surfaces and the dielectric material interface) and about the current carriers, whether they are electrons or ions [40,41].

From Figure 7, it is observed that the large values of the imaginary part ($Z^{\prime \prime }$) as compared to the real part ($Z^{\prime }$) of the complex impedance confirm a high capacitive behavior of these films, owing to their low conductivity [42]. Moreover, at low temperatures, all weight ratios of polymer blend samples exhibit nearly straight lines with large slope, suggesting the low conductivity of the samples. With increasing temperature, the curves bend toward the abscissa to form semicircular arcs, and the radii of semicircular arcs become smaller, indicating a higher value of conductivity at higher temperatures. This means that the hopping mechanism could be responsible for the electrical conduction in the present system [43].

The semicircle arc in the complex impedance plane plots are commonly used to estimate the *DC* bulk resistance $R_{b}$, of the dielectric material by extrapolating the intercept on the real axis *Z*′, then finding the $\sigma_{dc}$ values according to the following equation for all compositions [41]:(2)$$\sigma_{dc}=\frac{t}{R_{b}A}$$
where t (cm) is the thickness, A (cm^2^) is the electrode–electrolyte contact area, which is 3.14 cm^2^, and $R_{b}$ is the bulk resistance in ohms. 

The total conductivity $\sigma_{\omega }$ spectra at different temperature for all samples are shown in Figure 8, which are calculated by using the following relation [44]:(3)$$\sigma_{\omega }=\left[\frac{Z^{\prime }}{{Z^{\prime }}^{2}+{Z^{\prime \prime }}^{2}}\right]\left(\frac{t}{A}\right)$$

According to Figure 8, two distinct regions could be noted in the $\sigma_{\omega }$ spectra; a lower-frequency plateau region corresponding to DC conductivity, and a higher-frequency dispersion region corresponding to power law which has been defined by the Jonscher power law:(4)$$\sigma_{\omega }=\sigma_{dc}+\sigma_{ac}=\sigma_{dc}+D\omega^{s}\;\;\;\;\;\;$$

The first term ($\sigma_{dc})$ is the temperature-dependent (frequency-independent) DC conductivity and accounts for free charge resident in the bulk, while the second term ($\sigma_{ac})$ represents the frequency and temperature dependent ac conductivity and accounts for the bound and free charges. D is the dispersion parameter; ω is angular frequency (ω=2πf), and s is the power law exponent which generally varies between 0 and 1 [45].

The temperature dependence of conductivity for all complexes is in agreement with the theory founded by Croce et al. [46], and it is observed that the conductivity increased directly with temperature. This is explained due to the fact that the segmental motion causes an increase in the free volume of the system, which facilitates the movement of charge carriers [47,48]. It is well established that the improvement of conductivity with increasing temperature indicates the formation of voids presented by the amorphous area of the polymer blend electrolyte [49]. 

Increase in $\sigma_{ac}$ with frequency and temperature indicates that there may be charge carriers which are transported by hopping through the defect sites along the film structure [50]. It suggests that the blending PEO with MC is helpful in improving the electrical conductivity. This phenomenon is explained by investigation of the temperature-dependent DC conductivity $\left(\sigma_{dc}\right)$ for all compositions.

In this study, the value of $\sigma_{dc}$ has been attained by extrapolating of the plateau region of the total conductivity $\sigma_{\omega }$ in Figure 8 to the zero frequency (f→0). The obtained values of $\sigma_{dc}$ for all samples at various temperature ranges are presented in Table 2. According to Aziz et al. [24], the estimated $\sigma_{dc}$ from the plateau region of $\sigma_{\omega }$ spectra are comparable to the calculated $\sigma_{dc}$ values from the bulk resistance $R_{b}$. The room temperature DC conductivity for pure PEO and pure MC (Figure 9) are found to be respectively $13\times 10^{-10}\;S/cm$ and $1\times 10^{-10}\;S/cm$, which are compatible with the reported values in the literature [39,51,52]. The PEO:MC polymer blend with the weight ratio of 60:40 shows the highest room temperature DC conductivity of $6.55\times 10^{-9}\;S/cm$ and this value increased to $26\times 10^{-6}\;S/cm$ at 373 K. It is quite interesting to note that the greatest value of DC conductivity was associated with the minimum degree of crystallinity. This result has been confirmed by XRD and POM analysis. According to Buraidah et al. [53], increase in conductivity is due to more complexation sites provided by the blending of the two polymers, which is supported by the FTIR result. Hence, there will be more sites for carrier migration and exchange to take place in the amorphous phase of the polymer blend samples [54].

### 3.5. Dielectric Studies

Investigation of dielectric properties is another significant source of valuable information concerning electrical conduction processes; thus, the dielectric data for 60:40 weight ratio PEO:MC blend samples with different temperature can be analyzed using tangent loss (Dispersion factor D) tan δ, and complex electric modulus $M^{*}$. Figure 10 shows the variation of tangent loss $(\tan\;\delta \;=\varepsilon^{\prime \prime }/\varepsilon^{\prime }$) with frequency for PEO:MC blend sample with 60:40 weight ratio, at different temperatures between 303 and 373 K. 

The single relaxation peak of tan δ spectra appeared in Figure 10 can be used to determine the relaxation time ($\tau_{EP}=1/2\pi f_{EP}$), where $f_{EP}$ is the relaxation frequency [55]. The observed shift in the position of the relaxation peak towards the high-frequency side with increasing temperature indicates the decrease of relaxation time, which evidences the increase in electrical conductivity [56]. Thus, the dominant charge transport mechanism in these samples is mostly hopping of charge carriers among the trap levels situated in the band gap which can be described by the following equation [57]:(5)$$\sigma =\varepsilon_{o}\varepsilon^{\prime }\omega \tan\;\delta \;\;\;\;$$
where $\varepsilon_{o}$ is the permittivity for free space, $\varepsilon^{\prime }$ is the real part of dielectric constant, and ω is angular frequency (ω=2πf). Pradhan et al. [58] described that the peak shifting towards higher frequency causes reduction of the relaxation time due to an increase in carrier mobility. It is also obvious that the tan δ peak heights are increased due to increasing temperature. Parameswaran et al. [49] reported that the increase in the peak intensity indicates the breaking of bond formation from the dipoles. The peak observed in the tan δ spectra can also be attributed to the fact that the hopping frequency of the charge carriers is approximately equal to the frequency of the external applied field [59].

The mechanism of dielectric relaxation has been studied by the complex electric modulus ($M^{*}$), which is defined as the reciprocal of the complex permittivity ($\varepsilon^{*}$) as given by Equation (6). The advantage of this formulation is that the effects of electrode polarization are suppressed so that the electric modulus spectrum mainly reflects the bulk electrical properties of the samples [43]:(6)$$M^{*}=\;\frac{1}{\varepsilon^{*}}=M^{\prime }+jM^{\prime \prime }\;\;\;\;\;\;\;\;\;\;$$
where $M^{\prime }$ and $M^{\prime \prime }$ are, respectively, real and imaginary parts of the complex electric modulus. The frequency dependence of real $M^{\prime }$ and imaginary parts $M^{\prime \prime }$ of the electrical modulus at different temperatures for the highest conducting sample are presented in the Figure 11. Based on this plot, $M^{\prime }$ increased at the high-frequency end. This increasing trend at a higher frequency may be attributed to the bulk effect. At low frequency, $M^{\prime }$ approached zero, indicating that the contribution of electrode polarization is negligible [24]. On the other hand, the reduction in $M^{\prime }$ with increasing temperature is related to the segmental chain motion due to an increase in free volume caused by thermal expansion [60].

It is also evident from Figure 11 that the observed dispersion of $M^{\prime }$ at higher frequency is accompanied by a loss peak in the $M^{\prime \prime }$ spectra. The asymmetric broadening in the $M^{\prime \prime }$ peak indicates the presence of non-Debye-type relaxation behavior in the present sample. The peak of $M^{\prime }’$ can be further analyzed by using the relation shown in Equation (7). The change in peak position suggests temperature-dependent relaxation [61]; it is clearly seen that with increasing temperature, the $M^{\prime \prime }$ peak shifts towards higher frequency, but the peak height does not significantly change with temperature. This trend could be due to a decrease in the capacitance of the film.

The relaxation function φ(t), which describes the electric filed within the dielectric material, is related to relaxation time by the decay function, given as [62]:(7)$$\varphi \left(t\right)=\exp{\left(-\frac{t\;}{\tau_{m}}\;\right)}^{\beta }$$
where $\tau_{m}$ is the most probable relaxation time, and β=1.14/ω is the stretching exponent parameter. Here, ω is the full-width at half-maximum. The value of β is between 0 and 1, and for an ideal Debye relaxation, β=1. The smaller the value of β, the larger the deviation of relaxation with respect to Debye-type relaxation [63]. The value of β at different temperatures was calculated from the $M^{\prime \prime }$ spectra in Figure 11, and the values are tabulated in Table 3. The obtained values of β<1 reflect that the relaxations observed are a temperature-dependent non-Debye relaxation process [61].

## 4. Conclusions

The perfect composition for polymer blend films based on PEO and MC has been optimized using XRD, FTIR, POM, and EIS. The highest electrical conductivity (6.55 × 10^−9^ S/cm) at room temperature for the blended films was achieved for a sample consisting of 60% wt.% PEO and 40% wt.% MC. XRD and POM analysis confirm a reduction in the crystalline nature of this film. Thus, both techniques can be used for optimization of the composition of PEO polymer blends. According to the free-volume model, the increase of electrical conductivity of blend samples responds to the increment in the mobility of the charge carrier. From FTIR analysis, an interaction between PEO and MC is evidenced by the shift of the hydroxyl band to a lower wavenumber. Analysis of the complex electrical modulus and loss tangent (tan δ) shows that charge transport occurs via a hopping mechanism. The value of the stretching exponent parameter β < 1 reveals the presence of a non-Debye relaxation process in the present polymer blend system.

## Figures and Tables

**Figure 1 polymers-11-00853-f001:**
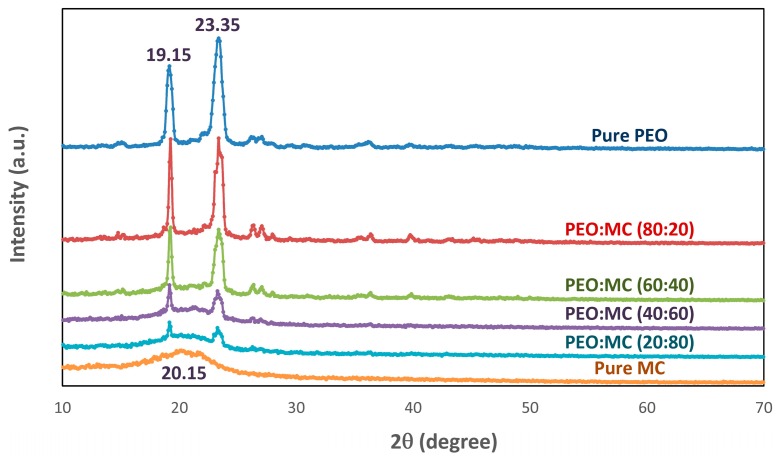
The X-ray diffraction pattern for pure polyethylene oxide (PEO), pure methylcellulose (MC) and their blend with different weight ratio.

**Figure 2 polymers-11-00853-f002:**
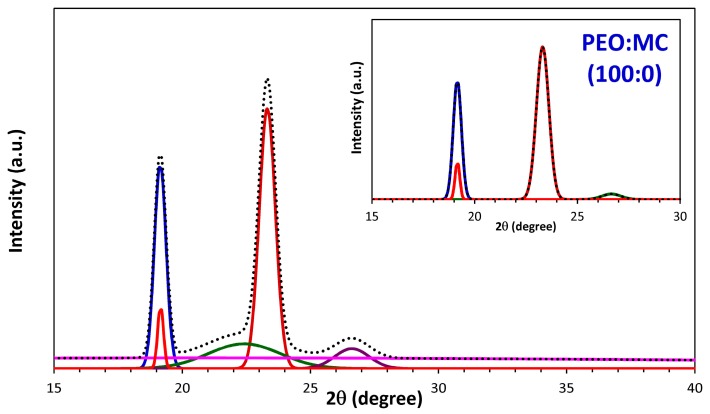
Deconvoluted XRD peaks for different concentrations of PEO:MC polymer blend. The inner graph depicts the peaks for the crystalline region, while the outer shows the peaks for both the crystalline and amorphous regions (dotted lines represent the sum of Gaussian peaks).

**Figure 3 polymers-11-00853-f003:**
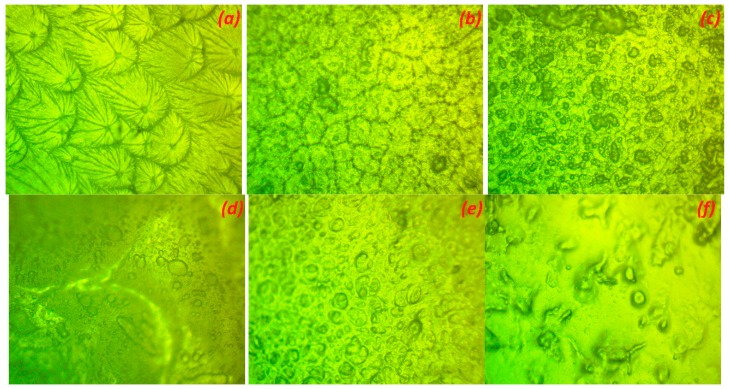
Optical micrographs of: (**a**) Pure PEO; (**b**) PEO:MC blend weight ratio 80:20; (**c**) PEO:MC blend weight ratio 60:40; (**d**) PEO:MC blend weight ratio 40:60; (**e**) PEO:MC blend weight ratio 20:80; (**f**) Pure MC.

**Figure 4 polymers-11-00853-f004:**
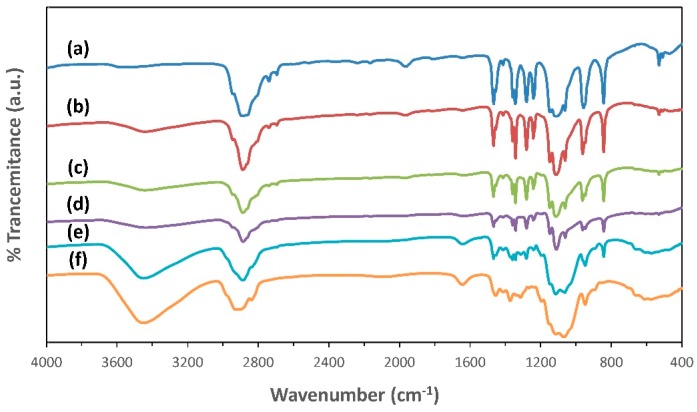
The FTIR spectra for: (a) Pure PEO; (b) PEO:MC blend weight ratio 80:20; (c) PEO:MC blend weight ratio 60:40; (d) PEO:MC blend weight ratio 40:60; (e) PEO:MC blend weight ratio 20:80; (f) Pure MC.

**Figure 5 polymers-11-00853-f005:**
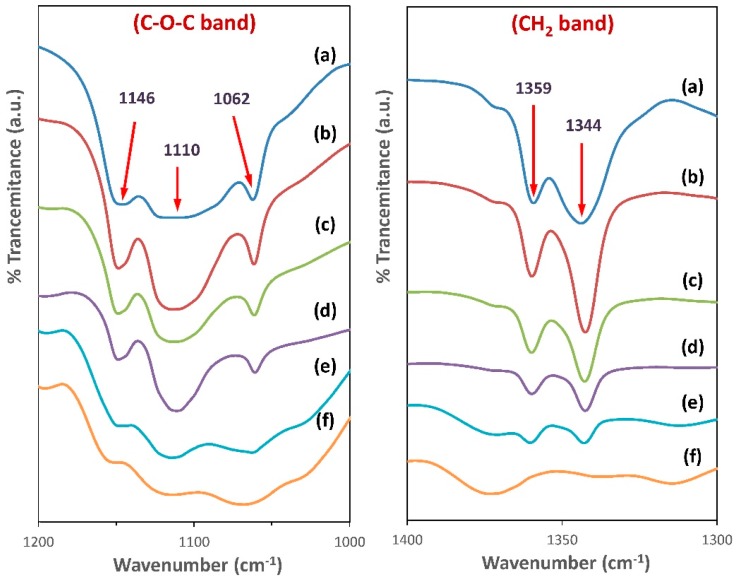
FTIR spectra in the regions 1000–1200 cm^−1^ and 1300–1400 cm^−1^ for: (a) Pure PEO; (b) PEO:MC blend weight ratio 80:20; (c) PEO:MC blend weight ratio 60:40; (d) PEO:MC blend weight ratio 40:60; (e) PEO:MC blend weight ratio 20:80; (f) Pure MC.

**Figure 6 polymers-11-00853-f006:**
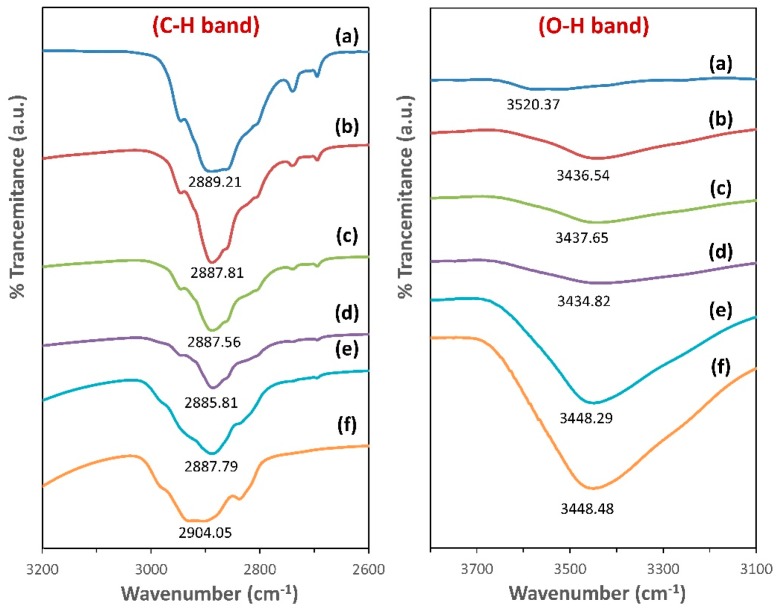
The stretching bond for C–H and O–H bands for: (a) Pure PEO; (b) PEO:MC blend weight ratio 80:20; (c) PEO:MC blend weight ratio 60:40; (d) PEO:MC blend weight ratio 40:60; (e) PEO:MC blend weight ratio 20:80; (f) Pure MC.

**Figure 7 polymers-11-00853-f007:**
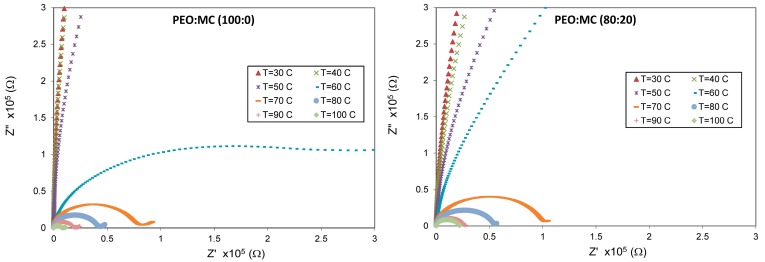
Nyquist impedance plots for PEO:MC films with different composition in the temperature range of 303–373 K.

**Figure 8 polymers-11-00853-f008:**
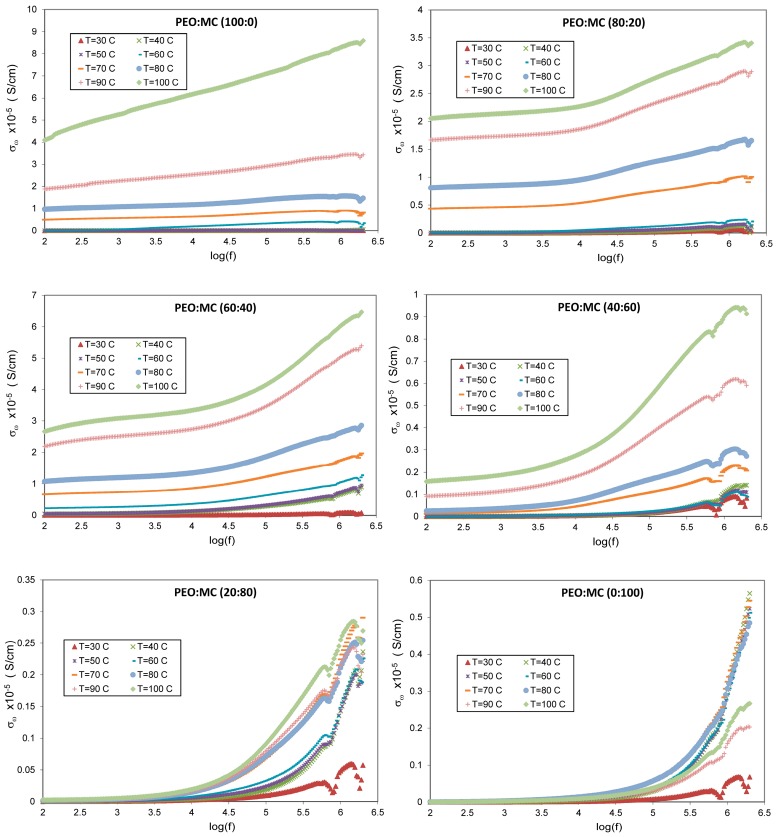
The total conductivity spectra for different compositions of PEO:MC in the temperature range of 303–373 K.

**Figure 9 polymers-11-00853-f009:**
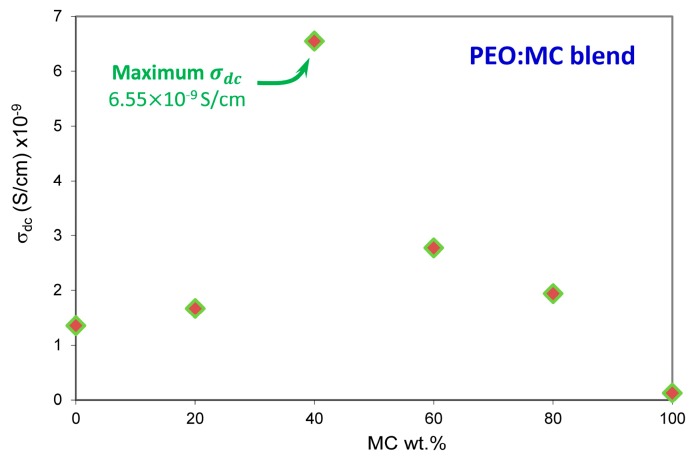
Variation in ambient temperature DC conductivities by MC wt.% in the PEO matrix.

**Figure 10 polymers-11-00853-f010:**
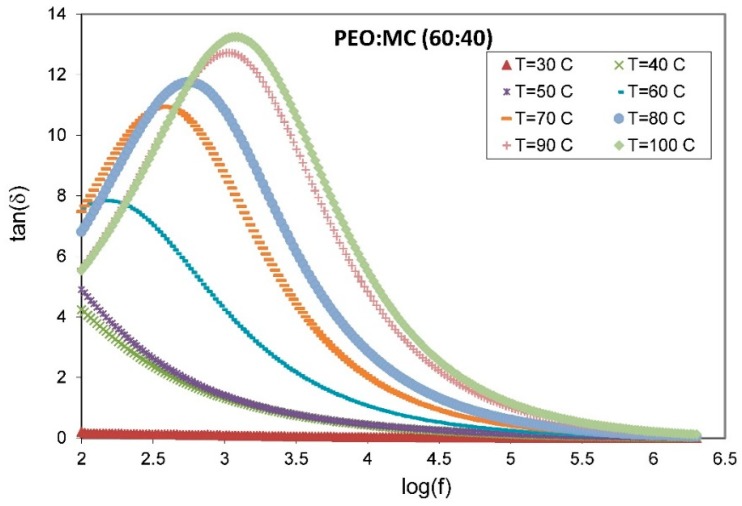
The variation of tan δ with frequency for the 60:40 weight ratio PEO:MC blend sample with at different temperatures.

**Figure 11 polymers-11-00853-f011:**
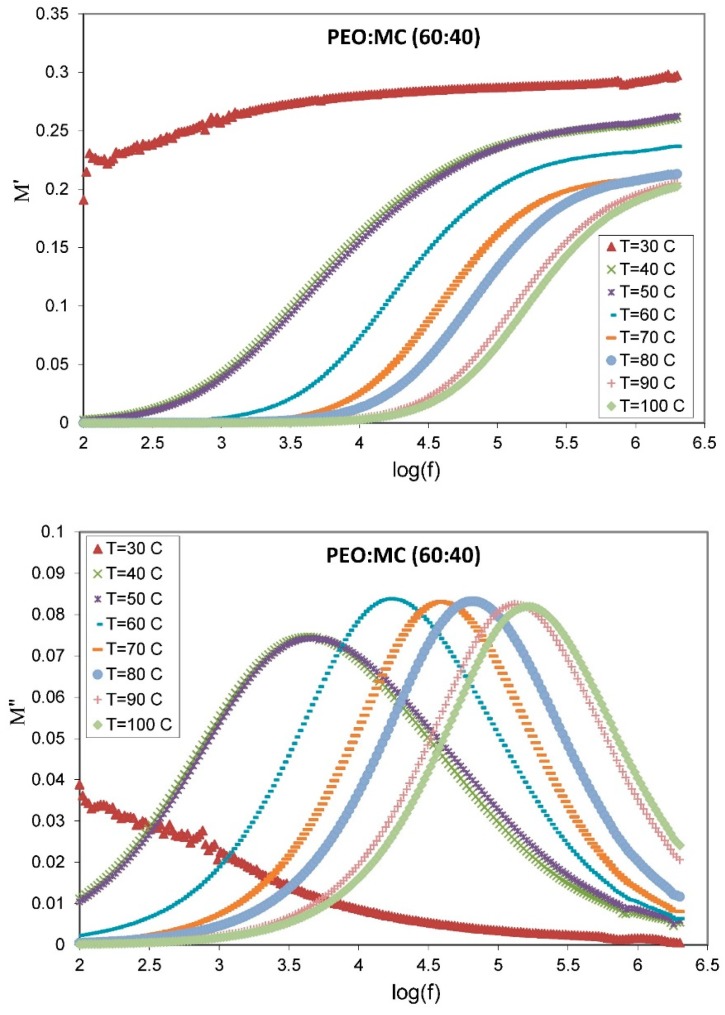
The frequency dependence of real $M^{\prime }$ and imaginary parts $M^{\prime \prime }$ of the complex electrical modulus at various temperatures from 303 to 373 K for 60:40 wt.% PEO:MC.

**Table 1 polymers-11-00853-t001:** Center, full width half maximum (FWHM) of the deconvoluted XRD peaks, and degree of crystallinity for PEO:MC polymer blend films with different compositions.

PEO:MC Ratio	First Peak		Second Peak		Third Peak		Forth Peak		Degree of Crystallinity %
	Center	FWHM	Center	FWHM	Center	FWHM	Center	FWHM	
100:0	23.311	0.714	19.153	0.470	26.649	1.105	---	---	32.33
80:20	19.233	0.252	23.383	0.589	26.303	0.263	27.041	0.252	21.48
60:40	19.209	0.267	23.373	0.604	26.603	1.040	---	---	15.86
40:60	19.161	0.228	23.317	0.625	20.507	4.176	26.942	1.657	16.37
20:80	20.165	4.337	23.308	0.610	19.162	0.213	---	---	16.81
0:100	20.786	5.173	19.656	4.619	21.594	1.858	---	---	16.10

**Table 2 polymers-11-00853-t002:** The DC conductivity at various temperature ranges (303–373 K) for different PEO:MC compositions.

PEO:MC Ratio	DC Conductivity (×10^−6^) (S/cm), at Different Temperatures							
	303 K	313 K	323 K	333 K	343 K	353 K	363 K	373 K
(100:0)	0.0013	0.0034	0.0103	0.1117	4.9682	9.6273	18.800	40.939
(80:20)	0.0016	0.0132	0.0324	0.1155	4.3675	8.0903	16.647	20.537
(60:40)	0.0065	0.4712	0.5109	2.2839	6.6989	10.748	21.898	26.677
(40:60)	0.0027	0.0018	0.0023	0.0039	0.1267	0.2457	0.9164	1.5767
(20:80)	0.0019	0.0016	0.0022	0.0020	0.0149	0.0169	0.0180	0.0249
(0:100)	0.0001	0.0019	0.0032	0.0029	0.0038	0.0050	0.0040	0.0063

**Table 3 polymers-11-00853-t003:** The values of FWHM and stretching exponent for $M^{\prime \prime }$ versus temperatures.

Temperature (K)	FWHM	β
313	2.169	0.525
323	2.078	0.548
333	1.760	0.647
343	1.684	0.676
353	1.480	0.770
363	1.454	0.784
373	1.428	0.798

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
