# Peer review of "Preparation and Composition Optimization of PEO:MC Polymer Blend Films to Enhance Electrical Conductivity"

_polymers, 2019, doi:10.3390/polym11050853_

Round 1
Reviewer 1 Report
In this manuscript by Omed Gh. Abdullah and Hawzhin T. Ahmed, the authors have demonstrated optimization of composition of a PEO:MC polymer blend. They have found that 60:40 ratio gives the best results in terms of enhancement of electrical conductivity of this material. Although the work is described at a resonably good level, I have a few suggestions, which could improve it:
1) Please improve the level of English. There are some errors present such as:
- Line 23 - DC conductivity, not dc conductivity
- Line 95 - to obtain, not to obtaining
- Line 200 - when different amount of MC was added, not when different amount of MC added
- etc.
2) Consider clearer presentation of the plots. Fig. 2 should be compressed to enable one to see all the results ideally on one page. Moreover, you should define in the caption the meaning of the dotted line, the insets and so on.
3) What regards all the figures, I suggest referring to the composition directly, so instead of using letters from "a" through "f" write explicitly pure PEO, 80:20 PEO:MC, and so forth. This would greatly simplify understanding of this results.
4) I suspect that Figures 6 to 8 were drawn using a different plotting application that the previous ones. Please try to make it coherent.
5) Fig. 9 is very worrying. No standard deviation bars suggest that only one measurement was taken for each sample. Please correct me if I am wrong. If not, please take more measurements to reach statistical significance. On another note, I would strongly suggest not to draw a line between these individual data points when you have not measured other compositions. This is misleading. Bar graph would probably be most informative in this place.
Other than that the work appears interesting and could be reconsidered for publication after taking into the consideration the aformenetioned corrections.
Author Response
Reviewer #1:
In this manuscript by Omed Gh. Abdullah and Hawzhin T. Ahmed, the authors have demonstrated optimization of composition of a PEO:MC polymer blend. They have found that 60:40 ratio gives the best results in terms of enhancement of electrical conductivity of this material. Although the work is described at a resonably good level, I have a few suggestions, which could improve it:
Dear Reviewer;
Thank you very much for your time and your very thoughtful set of constructive comments and suggestions regarding our paper. These comments have helped us a lot in clarifying the manuscript. Following your comments and suggestion, the manuscript was revised accordingly.
Below, we attempted to respond to your comments and recommendations. Please, kindly note that the major revised parts are highlighted in red color for your convenience of re-reviewing.
1) Please improve the level of English. There are some errors present such as:
- Line 23 - DC conductivity, not dc conductivity
- Line 95 - to obtain, not to obtaining
- Line 200 - when different amount of MC was added, not when different amount of MC added
- etc.
Thank you for these observations; All the corrections were made according to your suggestions. Moreover, all grammatical and typographical errors were corrected throughout the revised manuscript.
2) Consider clearer presentation of the plots. Fig. 2 should be compressed to enable one to see all the results ideally on one page. Moreover, you should define in the caption the meaning of the dotted line, the insets and so on.
Thank you for this comment; All corrections on Fig.2 were made according to your suggestions.
3) What regards all the figures, I suggest referring to the composition directly, so instead of using letters from "a" through "f" write explicitly pure PEO, 80:20 PEO:MC, and so forth. This would greatly simplify understanding of this results.
Dear respected reviewer, the letters were replaced by the composition ratio of polymer blend in Fig.1 and Fig. 2, to simplify understanding of this results as you suggested. Thank you for this suggestion.
4) I suspect that Figures 6 to 8 were drawn using a different plotting application that the previous ones. Please try to make it coherent.
Figures 7 and 8 are shown Nyquist impedance plots, and total conductivity spectra for all PEO:MC composition. They are plotted separately to show the effect of (1) composition, (2) frequency, and (3) temperature, on these parameters.
5) Fig. 9 is very worrying. No standard deviation bars suggest that only one measurement was taken for each sample. Please correct me if I am wrong. If not, please take more measurements to reach statistical significance. On another note, I would strongly suggest not to draw a line between these individual data points when you have not measured other compositions. This is misleading. Bar graph would probably be most informative in this place.
Thank you for this comment; Figure 9 was modified as you suggested. The respected reviewer might note that the range of investigated temperature (30-100 degree cellules) is beyond the glass transition temperature of PEO (67 degree cellules), and when polymer exceed its glass transition temperature it will undergo plastic deform. Thus for our samples the repeating measurement give different data which cannot be reliable.
Other than that the work appears interesting and could be reconsidered for publication after taking into the consideration the aformenetioned corrections.
Once again, we are grateful for the positive advice regarding our manuscript, which helped us make this a much better paper. We think that the manuscript has been greatly improved by these revisions, and we sincerely hope that our revised manuscript is now acceptable for publication.
Reviewer 2 Report
This paper describes the composition optimization of PEO/MC film through polymer blend to improve electrical conductivity. This article is well progressed with interesting, novel contents about enhancing electrical conductivity. Using diverse instruments, such as XRD, POM, FTIR, and EIS, the PEO/MC films were investigated at different component and different temperature, and the optimized film was founded. However, some information are ambiguous in the manuscript; thus, it would be recommended that the authors revise or adduce explanations.
The word “dc conductivity” should be changed to “DC conductivity”.
In fig. 2, there are two plots in each part (a to f). What is the difference between an outer plot and inner plot? It would be better to add this information.
It would be better to provide DSC data to explain the polymer blend.
In impedance analysis, an electrolyte was used, but there is no mention what material as an electrolyte.
The authors states that the greatest value of dc conductivity associated with the minimum degree of crystallinity (line 285 and 286). In table 2, only PEO film shows the highest Dc conductivity at 373K among the films despite high degree of crystallinity.
Author Response
Reviewer #2:
This paper describes the composition optimization of PEO/MC film through polymer blend to improve electrical conductivity. This article is well progressed with interesting, novel contents about enhancing electrical conductivity. Using diverse instruments, such as XRD, POM, FTIR, and EIS, the PEO/MC films were investigated at different component and different temperature, and the optimized film was founded. However, some information are ambiguous in the manuscript; thus, it would be recommended that the authors revise or adduce explanations.
Dear Reviewer;
Thank you very much for a very thoughtful set of constructive comments and suggestions regarding our paper. These comments have helped us a lot in clarifying the manuscript. Following your comments and suggestion, the manuscript was revised accordingly.
Below, we attempted to respond to your comments and recommendations. Please, kindly note that the major revised parts are highlighted in red color for your convenience of re-reviewing.
The word “dc conductivity” should be changed to “DC conductivity”.
Thank you for this comment; The word “dc conductivity” was changed to “DC conductivity”, throughout the revised manuscript.
In fig. 2, there are two plots in each part (a to f). What is the difference between an outer plot and inner plot? It would be better to add this information.
All the corrections on Fig.2 were made according to your suggestions.
It would be better to provide DSC data to explain the polymer blend.
We totally agree with the respected reviewer that DSC is one of the most important techniques in the study of polymeric materials. The reviewer might note that this study was focused on the composition optimization of polymer blend, which mostly perform by using EIS or XRD, in our study we showed that POM can also be used for this purpose.
In impedance analysis, an electrolyte was used, but there is no mention what material as an electrolyte.
Dear reviewer, as we mentioned above in this study we focused on the composition optimization of PEO:MC polymer blend, we do not use any salts.
The authors states that the greatest value of dc conductivity associated with the minimum degree of crystallinity (line 285 and 286). In table 2, only PEO film shows the highest Dc conductivity at 373K among the films despite high degree of crystallinity.
Thank you for this comment, it is well reported that the improvement of the electrical conductivity at ambient temperature is the only requirement for the materials to be used in electrochemical applications. Thus in this study we attempt to enhance the electrical conductivity at ambient temperature by reduction of crystallinity of PEO polymer. On the other hand, the investigation of electrical conduction as a function of frequency at different temperature aimed to understand conduction mechanism and nature of the charge transport predominant in this matrix.
Thank you once again for all your positive suggestions and comments to our paper, which helped us make this a much better paper. We sincerely hope that our revised paper is now acceptable for publication.
Reviewer 3 Report
An optimization in the composition of PEO:MC blend film in terms of its ionic conductivity is presented by the authors. Although systematic investigations were performed on the blend films of different composition, the current manuscript is like a technique report which lacks scientific significance. Therefore, I could not recommend the publication of this manuscript in its current form on a high-quality journal like Polymers. The following are suggestions for the authors for their further submissions.
1. The title of the manuscript can be revised. I don’t think polarized optical microscope is the main characterization that explains the changes in the conductivity of the films.
2. The authors should explain Why the peak center shift with changing composition. In addition, it appears to me that it is not convincing to conclude that PEO;MC (60:40) has a lowest crystallinity since the crystallinity for some other films are rather close.
3. The detailed experimental condition (humidity, thickness of each sample and etc.) for the EIS measurements should be described because the experimental condition significantly affects the impedance feature of the sample. In addition, a inset showing the high frequency behavior of the samples are preferred.
4. There are some typos in the manuscript (Line 362: bled; Line 231: tow).
Author Response
Reviewer #3:
An optimization in the composition of PEO:MC blend film in terms of its ionic conductivity is presented by the authors. Although systematic investigations were performed on the blend films of different composition, the current manuscript is like a technique report which lacks scientific significance. Therefore, I could not recommend the publication of this manuscript in its current form on a high-quality journal like Polymers. The following are suggestions for the authors for their further submissions.
Dear Reviewer;
We are really appreciated the task and efforts you put in reviewing our manuscript, These comments have helped us a lot in clarifying the manuscript. Following your comments and suggestion, the manuscript was revised accordingly.
Below, we attempted to respond your comments and recommendations. Please, kindly note that the major revised parts are highlighted in red color for your convenience of re-reviewing.
1. The title of the manuscript can be revised. I don’t think polarized optical microscope is the main characterization that explains the changes in the conductivity of the films.
Thank you for this comment, the respected reviewer might note that this study was focused on the composition optimization of polymer blend, which mostly perform by using EIS or XRD, in our study we showed that POM can also be used for this purpose.
2. The authors should explain Why the peak center shift with changing composition. In addition, it appears to me that it is not convincing to conclude that PEO;MC (60:40) has a lowest crystallinity since the crystallinity for some other films are rather close.
We totally agree with the respected reviewer that the change in crystallinity of blend polymer is small, this is because PEO in semi-crystalline nature, thus we cannot expect significant change upon adding amorphous polymer to semi-crystalline polymer. It has been well reported in the literature that the ion transport in polymer electrolyte is predominant through amorphous phase rather than the crystalline phase. Thus the reduction in crystallinity of this polymer blend is the first step for preparing polymer electrolytes with higher electrical conductivity at ambient temperature.
3. The detailed experimental condition (humidity, thickness of each sample and etc.) for the EIS measurements should be described because the experimental condition significantly affects the impedance feature of the sample. In addition, a inset showing the high frequency behavior of the samples are preferred
Thank you for this comment, all the required modifications in experimental part were performed following your suggestion. Thank you for these observations.
4. There are some typos in the manuscript (Line 362: bled; Line 231: tow).
Thank you for these observations; All the corrections were made according to your suggestions. Moreover, all grammatical and typographical errors were corrected throughout the revised manuscript.
Thank you once again for all your positive suggestions and comments to our paper, which helped us make this a much better paper. We sincerely hope that our revised paper is now acceptable for publication.
Reviewer 4 Report
Authors describes the composition optimization of PEO:MC polymer blend to enhance electrochemical conductivity. Unfortunately, reviewer cannot find the scientific importance of this paper. Especially, in the impedance analysis, what conductivity do authors measure? Proton conductivity or electron conductivity. They do not mention it at all. Moreover, authors don’t understand Croce et al [46] work at all. On the measurement of ionic conductivity for polymer electrolyte investigated by Croce et al, the polymer electrolyte contain some sort of salts. However, polymer electrolyte of this paper does not contain them. Why? Moreover, interpretation of IR date is questionable for me. The shift of the absorption peaks (see Fig. 6) is very small. It seems that it is difficult to discuss the interaction between PEO and MC. Therefore, this paper is not suitable for publication in Polymers.
Author Response
Reviewer #4:
Authors describes the composition optimization of PEO:MC polymer blend to enhance electrochemical conductivity. Unfortunately, reviewer cannot find the scientific importance of this paper. Especially, in the impedance analysis, what conductivity do authors measure? Proton conductivity or electron conductivity. They do not mention it at all. Moreover, authors don’t understand Croce et al [46] work at all. On the measurement of ionic conductivity for polymer electrolyte investigated by Croce et al, the polymer electrolyte contain some sort of salts. However, polymer electrolyte of this paper does not contain them. Why? Moreover, interpretation of IR date is questionable for me. The shift of the absorption peaks (see Fig. 6) is very small. It seems that it is difficult to discuss the interaction between PEO and MC. Therefore, this paper is not suitable for publication in Polymers.
Dear Reviewer;
We are appreciated your time and your efforts to reviewing our manuscript. Below, we attempted to respond your comments and recommendations. Please, kindly note that the major revised parts are highlighted in red color for your convenience of re-reviewing.
The reviewer might note that this study was focused on the composition optimization of polymer blend. Thus this study was an attempt to enhance the electrical conductivity at ambient temperature by reduction of crystallinity of PEO polymer. No salts were added to the samples; Our samples are “polymer blend” and not “polymer electrolyte”, this is why the shift of the FTIR peaks is small. There is some electrostatic interaction which caused the reduction in crystallinity of semi-crystalline PEO.
Thank you once again for your time to review our paper, we sincerely hope that our response to your feedback was satisfactory, and our revised paper is now acceptable for publication.
Round 2
Reviewer 1 Report
The authors have addressed most of my concerns, so I could recommend publication of this article. Nevertheless, if this work goes to another round of review, Figure 9 should really be changed according to my previous suggestions (remove the line between the individual data points and carry out more experiments to obtain a minimum level of statistical significance).
Author Response
Reviewer #1:
The authors have addressed most of my concerns, so I could recommend publication of this article. Nevertheless, if this work goes to another round of review, Figure 9 should really be changed according to my previous suggestions (remove the line between the individual data points and carry out more experiments to obtain a minimum level of statistical significance).
Dear Reviewer;
Thank you very much for your positive decision. Following your suggestion, the line between data points was removed in Figure 9. Thank you once again, for the positive advice regarding our manuscript, we appreciate it very highly.
Reviewer 2 Report
The authors revised the paper with contents required from reviewers. Nevertheless, many parts in the paper still seem to be ambiguous. In other words, there are little scientific explanations, especially, electrical conductivity.
In Fig. 2, compared to the previous version, the plots are moderately revised. The authors state that there are two regions, such as crystalline region and amorphous region, in outer plots. However, it still be not clear (i.e. which area is an amorphous region?).
In impedance data, what is the relation between a semi-circle and electrical conductivity for PEO:MC samples without salts?
In addition, only providing impedance data to demonstrate electrical conductivity is not convincing. In case of the electrical conductivity without salt, other measurements like four-point probe are more reliable.
The authors point out that POM, one of the measurements, can be used to prove the composition optimization of polymer blend. However, it is difficult to support their purpose as (c), (d), and (e) have a similar segregated domain. Besides, domain size can imply the phase behaviors for the samples. Due to no scale bar in the images, the scale of domain cannot be estimated.
Author Response
Reviewer #2:
The authors revised the paper with contents required from reviewers. Nevertheless, many parts in the paper still seem to be ambiguous. In other words, there are little scientific explanations, especially, electrical conductivity.
Dear Reviewer;
We are appreciating your time and your efforts in reviewing the revised version of our manuscript. Following your comments and suggestion, the manuscript was revised accordingly. Your feedback helped us a lot in clarifying the manuscript. Below, we attempted to respond your comments and recommendations. Please, kindly note that the major revised parts are highlighted in red color for your convenience of re-reviewing.
In Fig. 2, compared to the previous version, the plots are moderately revised. The authors state that there are two regions, such as crystalline region and amorphous region, in outer plots. However, it still be not clear (i.e. which area is an amorphous region?).
Thank you for this comment. Figure 2 is used to determine the degree of crystallinity for all composition of PEO:MC blend samples. The degree of crystallinity was calculated from the values of the area under crystalline peaks (Ac) and area under amorphous haloes (Aa), according to equation (1) in the manuscript.
The area under crystalline peaks (Ac) was calculated from inner plot of Fig. 2, while the outer plot shows the peaks for both crystalline and amorphous regions (Ac+Aa); from these two values, the crystalline degree can be calculated. Thus, we do not need to found the value of (Aa) alone in order to calculate (Xc). Nevertheless, the value of (Aa) can be obtained easily by subtracting (Ac) from (Ac+Aa).
In impedance data, what is the relation between a semi-circle and electrical conductivity for PEO:MC samples without salts?
It is well known by respected reviewer, that the impedance plots in general exhibit two noticeable regions: the high-frequency semi-circular arc and low-frequency tail. For perfect Debye-type relaxation, a complete semicircle with its centre overlaps with the Z' axis should be observed, whereas the deviation from Debye-type behaviour was expected by detecting a depressed semicircle whose centers are below Z' axis. The complex impedance plane plots are commonly used to separate the bulk material (at a higher frequency) and the electrode surface polarization (at low-frequency) phenomena. Thus, the semicircle arcs in the complex impedance plane plots are commonly used to estimate the DC bulk resistance of the dielectric material by extrapolate intercept on the real axes Z′ and finding the DC conductivity values according to equation number (2) in the revised manuscript. The authors would like to appreciate the reviewer for this comment; The paragraphs were revised to clarify the description of impedance plot and electrical conductivity, as you suggested.
In addition, only providing impedance data to demonstrate electrical conductivity is not convincing. In case of the electrical conductivity without salt, other measurements like four-point probe are more reliable.
We agree with the respected reviewer that the four-point probe has proven to be a convenient tool, with a simple apparatus for measuring the electrical conductivity of conducting or semiconductor materials. In this study we found the electrical conductivity as a function of composition, frequency, and temperature, to understand a conduction mechanism to tuning its value by controlling the compositions (the focus of this study).
The authors point out that POM, one of the measurements, can be used to prove the composition optimization of polymer blend. However, it is difficult to support their purpose as (c), (d), and (e) have a similar segregated domain. Besides, domain size can imply the phase behaviors for the samples. Due to no scale bar in the images, the scale of domain cannot be estimated.
We also agree with the reviewer’ view point, that the samples (c), (d), and (e) have almost similar segregations. In the POM figures, the dark regions attributed to the amorphous domain, and the observed spherulitic represent the crystalline domain. Upon blending the size of spherulitics were reduced and the enhancement in the dark region was observed. According to XRD analysis, and the data shown in Table 1, the degree of crystallinity of the samples (c), (d), and (e) is respectively 15.8, 16.3 and 16.8, i.e., the change in the crystallinity of these samples is too small. Thus we cannot expect a significant change in the POM figures for these samples, thus they seem to be relatively similar. Following this comment, the title of the manuscript has been revised to a more precision title. Finally, about the scale bare all the figure have the same magnification of 10x, as mentioned in the "Sample characterization" section.
Thank you once again for your time to review our revised paper, we sincerely hope that our response to your feedback was satisfactory, and our revised paper is now acceptable for publication.
Reviewer 3 Report
The authors have addresed most of my concerns and I suggest acceptance of the revised manuscript.
Author Response
Reviewer #3:
The authors have addressed most of my concerns and I suggest acceptance of the revised manuscript.
Dear Reviewer;
Thank you so much for your positive decision, we appreciate it very highly.
Reviewer 4 Report
Authors claim that increase of amorphous region of PEO might be achieved by MC blending method into PEO in this paper. The polymer bended methods into semi-crystalline PEO to increse amorphous region are wellknown in large research reports of polymer electrolytes. Are there any novel sicientific findings except for utilization of MC? Please mention this points in the introduction. Morover, reviewer has two comments below.
1) Both PEO and MC are soluble in water. Authors obtained the films by drying samples on the silica gel and at ambient temperature. How do authors confirm that samples are completely dry state? Hydrogen bond between water and PEO and MC is also present in films. And also, they use large amount of water to dissolve the PEO and MC. It is difficult to remove water completly from the films by authors' treatment. The presence of water might affect the physical properties of the films as a plasticizer. Does the water present in films affect in all measurements carried out in this work?
2) Degree of crystallinity of the polymer blend is calculated and the numbers are shown in Table 1. Reviewer does not think that the level of the precision might be expressed as 3 digits after the number of decimal points though it is possible to show the numbers obtained by calculation. Need to revise the numbers.
Author Response
Reviewer #4:
Authors claim that increase of amorphous region of PEO might be achieved by MC blending method into PEO in this paper. The polymer bended methods into semi-crystalline PEO to increase amorphous region are wellknown in large research reports of polymer electrolytes. Are there any novel scientific findings except for utilization of MC? Please mention these points in the introduction. Moreover, reviewer has two comments below.
Dear Reviewer;
Thank you very much for a very thoughtful set of constructive comments and suggestions regarding our paper. These comments have helped us a lot in clarifying the manuscript. Following your suggestion, the introduction part was revised accordingly.
1) Both PEO and MC are soluble in water. Authors obtained the films by drying samples on the silica gel and at ambient temperature. How do authors confirm that samples are completely dry state? Hydrogen bond between water and PEO and MC is also present in films. And also, they use large amount of water to dissolve the PEO and MC. It is difficult to remove water completely from the films by authors' treatment. The presence of water might affect the physical properties of the films as a plasticizer. Does the water present in films affect in all measurements carried out in this work?
The authors would like to thank the reviewer for precise and this thoughtful comment. We agree with the comment, absolutely the presence of H2O will affect all the physical properties of the film. Although prior to characterization, the films were kept in a desiccator with blue silica gel for further drying, the presence of water molecules was observed in FTIR spectra. This can be noted from the broadness of the OH- band of the samples. This is why we say, the preparation technique affects the physical properties of the prepared films.
2) Degree of crystallinity of the polymer blend is calculated and the numbers are shown in Table 1. Reviewer does not think that the level of the precision might be expressed as 3 digits after the number of decimal points though it is possible to show the numbers obtained by calculation. Need to revise the numbers.
Thank you for this comment. The appeared numbers in Table 1, modified according to your suggestion.
Thank you once again for all your positive suggestions and comments to our paper, which helped us, makes this a much better paper. We sincerely hope that our response to your feedback was satisfactory, and our revised paper is now acceptable for publication.
Round 3
Reviewer 2 Report
The authors revised the manuscript so that the ambiguous information and statements were removed. Due to their endeavor, the article named “Preparation and composition optimization of PEO:MC polymer blend films to enhance electrical conductivity” now has good quality. In other words, this article contains scientific information and significant results regarding electrical conductivity in the field of the polymer blend. Using diverse measurements, such as XRD, FT-IR, and POM, the authors provide the properties of PEO:MC blend at different weight ratios. In addition, especially, to investigate the electrical conductivity of the polymer blend films, the measurement of EIS was used, which supports the authors’ statement. I think the article is valuable to be involved in the journal, Polymers.